# The Underlying Mechanisms of Curcumin Inhibition of Hyperglycemia and Hyperlipidemia in Rats Fed a High-Fat Diet Combined With STZ Treatment

**DOI:** 10.3390/molecules25020271

**Published:** 2020-01-09

**Authors:** Zhen-Hong Xia, Wen-Bo Chen, Li Shi, Xue Jiang, Ke Li, Yu-Xiang Wang, Yan-Qiang Liu

**Affiliations:** College of Life Sciences, Nankai University, Tianjin 300071, China; xiazhenhong22@sina.com (Z.-H.X.); 1120180396@mail.nankai.edu.cn (W.-B.C.); darksleepinger@gmail.com (L.S.); jxmagic0626@163.com (X.J.); taxono@163.com (K.L.); wangyuxiang1115@126.com (Y.-X.W.)

**Keywords:** anti-apoptosis, anti-oxidant, curcumin, hyperglycemia, hyperlipidemia

## Abstract

Curcumin is the main secondary metabolite of *Curcuma longa* and other *Curcuma* spp, and has been reported to have some potential in preventing and treating some physiological disorders. This study investigated the effect of curcumin in inhibiting high-fat diet and streptozotocin (STZ)-induced hyperglycemia and hyperlipidemia in rats. Twenty-six male Sprague-Dawley (SD) rats (170–190 g) were randomly divided into a standard food pellet diet group (Control group), a high-fat diet and streptozotocin group (HF + STZ group), and a high-fat diet combined with curcumin and STZ group (HF + Cur + STZ group). Compared with the HF + STZ group, the HF + Cur + STZ group exhibited significantly reduced fasting blood glucose (FBG), total cholesterol (TC), triglyceride (TG), low-density lipoprotein cholesterol (LDL-C), high-density lipoprotein cholesterol (HDL-C), alanine aminotransferase (AST), and aspartate transaminase (ALT) levels, as well as liver coefficients. In the livers of these rats, the expression of malondialdehyde (MDA) and Bax was downregulated, whereas that of superoxide dismutase (SOD) and Bcl-2 was upregulated. Moreover, the liver histology of these rats was improved and resembled that of the control rats. These results suggest that curcumin prevents high-fat diet and STZ-induced hyperglycemia and hyperlipidemia, mainly via anti-oxidant and anti-apoptotic mechanisms in the liver.

## 1. Introduction

Diabetes mellitus (DM) is common endocrine disease involving metabolic disorders on a global scale [1]. DM is characterized by hyperglycemia due to relative or absolute deficiencies in insulin action or insulin secretion [2]. According to the 2016 Global Report on Diabetes of the World Health Organization (WHO), 422 million people (or 8.5% of the population) were suffering from DM globally. DM and its complications affect the quality of life of human beings, and have become a significant global public health problem [3]. Because of the continuous increase in the numbers of patients and aggravation in prevalence, DM is one of the major chronic noninfectious diseases worldwide, and has imposed a heavy burden on human beings [3]. Thus, research and development of effective medicine for controlling DM and its complications is of great significance.

Lipid accumulation in liver tissue is closely related to type 2 diabetes [4]. As an important metabolic organ, the liver participates in almost all metabolic processes in the body and is the main organ responsible for regulating energy balance [5,6]. The liver is the main site of glucose and fatty acid metabolism and plays an important role in glycolipid metabolism. In the fasted state, the liver can maintain the sugar balance in the blood by enhancing gluconeogenesis and glycogenolysis; in the fed state, the liver can store energy through glycogen and fatty acid synthetic pathways [7]. The glycogen, triglyceride, and cholesterol levels in the liver are important indicators of lipid balance [8]. At the 2001 American Diabetes Society Annual Meeting, McGarry proposed the concept of diabetes mellitus, suggesting that patients with type 2 diabetes have abnormal lipid metabolism and that triglyceride deposition in the muscles precedes the abnormality in glucose metabolism [8]. Under the stimulation of long-term high-fat and high-sugar diets, the metabolic function of the liver is impaired, and the morphological structure is damaged to some extent [4]. This has close relationship with blood glucose and blood triglycerides. High blood glucose can cause the metabolic disorder of both glucose and lipids [9]. High blood triglycerides are one of the main indexes of hyperlipidemia, can influence glucose metabolism, and increase blood glucose [9]. In addition, several in vitro studies, as well as human experiments, have demonstrated negative effects of elevated levels of lipids and glucose on β-cell function in healthy and diabetic individuals, known as lipotoxicity and glucotoxicity [9]. Higher glucose levels shift the intercellular fatty acid metabolism to cellular lipid synthesis, leading to cytosolic accumulation of lipid-derived signaling molecules [10].

The liver is the most important organ involved in glucose and lipid metabolism, as well in as the development of diabetes mellitus [11,12]. Liver disease might cause diabetes mellitus; diabetes can also cause liver damage. Hyperglycemia is one of the main characteristics of diabetes mellitus, and can induce the increase of liver apoptosis by damaging the mitochondrial structure of liver cells [11,12]. The Bcl-2 protein family plays an important role in regulating apoptosis, whereby some proteins such as Bcl-2 are anti-apoptotic, while others such as Bax are pro-apoptotic [13]. Cells, including liver cell apoptosis, are regulated by both the *Bax* and *Bcl-2* genes, and the ratio of the *Bax* and *Bcl-2* genes might determine apoptosis status [14].

Curcumin (1, 7-bis (4- hydroxy- 3- methoxyphenyl)-1, 6- heptadiene-3, 5- dione) is a yellow phenolic compound present in turmeric (*Curcuma longa*), a widely used spice in Indian cuisine. [15]. Its chemical structure is composed of two O-methylated phenols and a beta-diketone, with a molecular formula of C_21_H_20_O_6_ and a molecular weight of 368.39 [16]. Because its phenolic hydroxyl group can directly capture or scavenge free radicals, curcumin is a natural phenolic antioxidant [16]. Curcumin has a strong ability to scavenge superoxide anions, and it is also an inducible nitric oxide synthase (iNOS) inhibitor [17,18]. Curcumin is used as a flavoring agent and food colorant, and has been found to have many pharmacological properties such as anti-inflammatory, anti-lipid peroxidation, anti-tumor, and liver protection effects, among others [19]. It has been reported to have some potential in preventing and treating cancer, diabetes and its complications, autoimmune diseases, atherosclerosis, arthritis, stroke, peripheral neuropathy, and enteritis [20,21,22,23,24]. In addition, curcumin has also exhibited pharmacological potential in preventing and treating central nervous system diseases such as Alzheimer’s disease (AD), Parkinson’s disease (PD), epilepsy, cerebral ischemia, and so on [25,26,27]. Although one study suggests that curcumin has some potential in controlling diabetes, its mechanism in controlling diabetes is unknown [28].

In this study, we observed the effect of curcumin pretreatment on body weight, blood glucose, organ coefficient, and other biochemical parameters in the blood such as total cholesterol (TC), triglyceride (TG), high-density lipoprotein cholesterol (HDL-C), low-density lipoprotein cholesterol (LDL-C), alanine aminotransferase (AST), aspartate transaminase (ALT), total protein (TP), and globulin (GLB). Further, we also evaluated oxidative stress markers and apoptosis-related protein expression in the liver, and performed a histological evaluation of the liver in all experimental animals, so as to understand the role of curcumin pretreatment on glycolipid metabolism in a high-fat diet combined with STZ treatment-induced rat model, and the underlying mechanisms. We hope to provide an experimental basis for administering curcumin to prevent and control hyperglycemia and hyperlipidemia and the development of diabetes.

## 2. Results

### 2.1. The Effect of Curcumin on Glucose Tolerance in Rats Fed A High-Fat Diet

In the oral glucose tolerance test (OGTT), the area under the glucose curve of the high-fat diet and streptozotocin (HF + STZ) group was significantly increased compared with that in the control group (*p* < 0.01), while the area under the glucose curve of the high-fat diet combined with curcumin and STZ group (HF + Cur + STZ) group was significantly decreased compared with that for the HF + STZ group (*p* < 0.01), as shown in Figure 1.

### 2.2. The Effect of Curcumin on Bodyweight and Fasting Blood Glucose (FBG) in Rats Fed a High-Fat Diet Combined with STZ Treatment

The bodyweights of each group showed an increasing trend before the injection of STZ. In the 12th week, the bodyweights of the rats in the HF + STZ group sharply decreased after STZ injection. The weight of the HF + Cur + STZ group was also decreased, as shown in Figure 2A.

One week after STZ injection, the rats were fasted for 12 h to measure FBG levels. As shown in Figure 2B, the mean FBG level in the HF + STZ group was significantly higher than that in the control group (*p* < 0.01); compared with the HF + STZ group, the FBG levels in the HF + Cur + STZ group were significantly decreased (*p* < 0.01).

Seven out of ten rats in the HF + STZ group were successfully developed as a diabetic model. Ten rats in the HF + Cur + STZ group did not develop hyperglycemia, indicating that there was no diabetic model.

### 2.3. The Effect of Curcumin on the Coefficients of the Liver to Body Weight of Rats Fed A High-Fat Diet Combined with STZ Treatment

The coefficients of the liver to body weight are expressed as milligrams (wet weight of tissues)/grams (fasted body weight). As shown in Figure 3, compared with the control group, the coefficients of the liver in the HF + STZ group were significantly increased (*p* < 0.01). Compared with the HF + STZ group, the coefficients of the liver in the HF+ Cur + STZ group were significantly decreased (*p* < 0.05), indicating that curcumin intervention reversed the increase caused by a high-fat diet combined with STZ treatment.

### 2.4. The Effect of Curcumin on Biochemical Blood Indicators in Rats Fed A High-Fat Diet Combined with STZ Treatment

The effects of curcumin on lipid metabolism are shown in Figure 4. Compared with the control group, the blood total cholesterol (TC), triglyceride (TG), low-density lipoprotein cholesterol (LDL-C), and high-density lipoprotein cholesterol (HDL-C) values in the HF + STZ group were significantly increased (*p* < 0.05). Compared with the HF + STZ group, the blood TC, TG, LDL-C, and HDL-C values in the HF +Cur + STZ group were significantly decreased (*p* < 0.05). The effect of curcumin on the biochemical index of liver function is shown in Figure 5. Compared with the control group, the levels of blood aspartate transaminase (ALT) and alanine aminotransferase (AST) in the HF + STZ group were significantly increased (*p* < 0.05). Compared with the HF + STZ group, the blood ALT and AST levels in the HF + Cur + STZ group were significantly downregulated (*p* < 0.05). Further, compared with the control group, blood total protein (TP) and globulin (GLB) levels in the HF + STZ group were significantly decreased (*p* < 0.05). Compared with the HF + STZ group, the blood TP and GLB levels in the HF +Cur +STZ group were increased to a certain extent, but the difference was not statistically significant.

### 2.5. The Effect of Curcumin on Oxidative Stress Markers in Liver Tissues of Rats Fed A High-Fat Diet Combined with STZ Treatment

As shown in Figure 6, the levels of malondialdehyde (MDA) in the liver of the HF + STZ group were significantly upregulated compared with those in the control group (*p* < 0.01), while the levels of MDA in the livers of the HF + Cur + STZ group were significantly downregulated compared with those in the HF + STZ group (*p* < 0.01). In addition, compared with the control group, the levels of superoxide dismutase (SOD) in the livers of the HF + STZ group were significantly downregulated (*p* < 0.05), while the SOD levels in the livers of the HF + Cur + STZ group were significantly upregulated (*p* < 0.05) compared with those in the HF + STZ group.

### 2.6. The Effect of Curcumin Intervention on Bcl-2 and Bax Expression in Rats Fed A High-Fat Diet Combined with STZ Treatment

Figure 7A shown that compared with the control group, the HF + STZ group exhibited reduced Bcl-2 expression, whereas curcumin intervention reversed the decrease caused by HF + STZ in the expression of Bcl-2 to some degree. Further analysis showed that compared with the control group, the HF + STZ group had significantly decreased average optical densities, reflecting decreased Bcl-2 expression (*p* < 0.05), as shown in Figure 7B. However, the average optical densities in the HF + Cur + STZ group were significantly higher than those in the HF + STZ group (*p* < 0.05). These results demonstrate that curcumin intervention can prevent a decrease in Bcl-2 expression in rats fed a high-fat diet combined with STZ treatment. Figure 7C shows that compared with the control, the HF + STZ group exhibited increased Bax expression, whereas curcumin intervention could reverse the increase in Bax expression caused by HF + STZ treatment to some degree. Further analysis showed that compared with the control group, the HF + STZ group had significantly increased average optical densities, representing the relative value of Bax expression (*p* < 0.01) (Figure 7D). However, the average light density in the HF + Cur +STZ group was significantly lower than that in the HF + STZ group (*p* < 0.05). These results demonstrate that curcumin intervention can prevent the increased Bax expression in rats fed a high-fat diet combined with STZ treatment.

### 2.7. The Effect of Curcumin Treatment on Liver Histopathology in Rats Fed A High-Fat Diet Combined with STZ Treatment

Histopathological changes in the liver tissue of various groups are summarized in Figure 8. Micro examination of liver structures showed that in the control group, the hepatocytes were uniform in size and the structure of hepatic lobules was clear. The hepatic cords were arranged neatly from the central vein to the surrounding area, and the arterioles and venules in the portal area were normal. The liver structures of rats in the HF + STZ group were destroyed, in which many hepatocytes showed necrosis, with a loss of their normal shape and architecture, cytoplasmic vacuolation, and missing or pyknotic nuclei. In addition, the sinusoids in the livers were dilated. There was fatty deposition and the leukocyte infiltration in the liver. However, compared with the HF + STZ group, the histopathological changes in the liver tissue in HF + Cur + STZ group rats were improved to some extent, showing significantly reduced infiltration of inflammatory cells, neatly arranged cell lines.

## 3. Discussion

In this study, we investigated the inhibition of a high-fat diet with streptozotocin (STZ)-induced hyperglycemia and hyperlipidemia in rats by curcumin. Our main finding was that curcumin decreased fasting blood glucose, blood TC, TG, LDL-C, HDL-C, ALT, and AST levels, as well as liver MDA levels and Bax protein, but upregulated liver SOD levels and Bcl-2 protein while improving liver microstructures in high-fat diet and STZ-treated rats. These results suggest that curcumin prevents high-fat diet and STZ-induced hyperglycemia and hyperlipidemia via anti-oxidant and anti-apoptotic mechanisms in the liver.

Glucose tolerance refers to the body’s ability to regulate blood sugar concentration, which is used as an index in the clinical diagnosis of diabetes mellitus [29]. Under normal circumstances, the body has a strong tolerance to glucose, while in type 2 diabetes, the body’s insulin receptor is not sensitive, and as such glucose tolerance is decreased [30]. The area under the blood glucose curve decreases significantly, reflecting the increased tolerance to glucose [31]. In the present study, we observed that the area under the blood glucose curve was decreased after treatment with curcumin, which suggested that curcumin increases tolerance to glucose in rats fed a high-fat diet. Some studies suggest that modulation of glucose tolerance are involved in gut macrobiotics [32] and the insulin signaling path [33]. Direct experimental evidence showed that curcumin improves glucose tolerance via stimulation of glucagon-like peptide-1 secretion [34]. Therefore, curcumin increasing tolerance to glucose in rats fed a high-fat diet might be involved in gut macrobiotics and insulin signaling path including mediation of glucagon-like peptide-1.

The organ index is an index reflecting the relative growth of organs. To some extent, it can indicate the function of organs [35]. Studies have observed that liver weights in animals treated with eugenol were reduced compared to that in the control animals [36]. The results of the present study showed that the liver index increased after high-fat feeding combined with STZ treatment, while the liver index in the curcumin intervention group was not significantly different from that of the control group, which might indicate that high-fat feeding combined with STZ treatment made the liver swell to some extent, and curcumin can inhibit swelling of the liver induced by a high-fat combined with STZ. Micro examination results of liver structures might further support that curcumin has protective effects on the liver of rat administered high-fat feeding combined with STZ treatment.

Blood glucose, triglycerides, and total cholesterol can reflect the metabolism of carbohydrates and fats in animals or humans. Hyperlipidemia is caused by disorders of lipid metabolism or abnormal transport in the fasting state, and shows no less than one higher-than-normal blood levels of TC, TG, and LDL-C, often accompanied by decreased blood HDL-C [37]. Long-term dyslipidemia can lead to a series of complications, such as fatty liver and atherosclerosis [38]. Several in vitro studies, as well as human experiments, have demonstrated negative effects of elevated levels of lipids and glucose on β-cell function in healthy and diabetic individuals, known as lipotoxicity and glucotoxicity, respectively [9]. Higher glucose levels shift the intercellular fatty acid metabolism to cellular lipid synthesis, leading to cytosolic accumulation of lipid-derived signaling molecules [10]. Diabetes is closely related to blood lipid levels, and lowering blood lipid levels can help improve diabetes [39]. In our study, the levels of blood TC, TG, LDL-C, and HDL-C in rats fed a high-lipid diet and then treated with STZ were significantly increased; curcumin pretreatment reversed these changes in blood TC, TG, LDL-C, and HDL-C. These findings suggest that a high-lipid diet followed by STZ treatment causes lipid metabolism disorders in rats, and curcumin intervention effectively improves the lipid metabolism disorder. These findings are similar to the result of previous in vitro experiment, in which curcumin decreased glycerol release in differentiated 3T3-L1 adipocytes, as it has a certain effect on regulating lipid metabolism dysfunction [40]. Generally, HDL-C could stimulate cell growth and tissue regeneration, and is helpful for the prevention of oxidant stress-disease including atherosclerosis and coronary heart disease [41,42]. However, in our study, HF + STZ treatment resulted in the highest blood HDL-C level in rats. A study reported a similar result in that compared with the rats in the normal group, the level of HDL-C in type 2 diabetes mellitus rats was obviously increased [3]. It may be the HDL-C increased responsiveness to high blood glucose status; on the issue further research is warranted.

It is well-known that the liver plays a crucial role in maintaining blood glucose homeostasis, and liver dysfunction is a significant cause of death in patients with T2DM [43,44]. The liver releases glucose based on the metabolic needs and is an important organ for carbohydrate metabolic homeostasis [45]. Recently, liver damage has been documented as a major complication of DM [46]. Indeed, various studies suggest that mortality due to liver disease in diabetic patients is very high, even higher than that for cardiovascular diseases [47]. While AST, ALT, TP and GLB are important indicators of liver function, in our study we found that AST and ALT levels were elevated in rats fed a high-lipid diet then those treated with STZ. However, TP and GLB levels were decreased, while curcumin intervention can reverse these changes, suggesting that the high-lipid diet in combination with STZ causes functional liver damage in rats, and curcumin intervention can reverse this damage.

Curcumin has been found to possess many pharmacological effects such as anti-inflammatory, anti-lipid peroxidation, anti-tumor, and liver protection activities, among others [19]. Curcumin can prevent experimental diabetic retinopathy in rats through its hypoglycemic, antioxidant, and anti-inflammatory effects [48]. Research has shown that curcumin improves liver injury and fibrosis [15]. The protective effects of curcumin against ischemia–reperfusion injury in tissues such as the lungs, cardiomyocytes, and liver have been previously reported [49,50]. Some studies have shown that curcumin protects astrocytes from oxidative stress, thus potentially using them for treating various neurodegenerative diseases [51]. In addition, DM is highly correlated with liver inflammation, cirrhosis, apoptosis, and ultimately causes liver malfunction [52].

Oxidative stress is involved in the pathogenesis of numerous diseases and is associated with an increase in lipid peroxidation, as indicated by higher concentrations of MDA [13,53], which is one of several low-molecular-weight end products formed via the decomposition of certain primary and secondary lipid peroxidation products [54]. Therefore, the levels of MDA imply the degree of lipid peroxidation, and indirectly reflect the level of oxidative stress. The results of our experiment showed that a high-fat diet combined with streptozotocin treatment significantly upregulated MDA levels and significantly downregulated SOD levels in experimental rats. Curcumin pretreatment reversed the changes in MDA and SOD induced by a high fat-diet combined with streptozotocin treatment to some extent, suggesting that a high-fat diet and streptozotocin treatment cause oxidative stress in the liver. Curcumin pretreatment inhibited the oxidative stress caused by a high-fat diet combined with streptozotocin treatment.

The Bcl-2 protein family plays an important role in regulating apoptosis [14]. In the Bcl-2 protein family, some proteins, such as Bcl-2 and Bcl-XL are anti-apoptotic, while others such as Bax and Bcl-XS are pro-apoptotic [55]. Generally, apoptosis is regulated by both Bax and Bcl-2 genes, and the ratio of *Bax* and *Bcl-2* genes reflects apoptosis status to some extent [56]. In this study, we found that expression of the Bax protein in liver tissues of rats fed a high-lipid diet and then treated with STZ was increased, while the expression of the Bcl-2 protein was decreased. However, curcumin treatment reversed these changes to some extent. Thus, we suggest that curcumin improves the apoptotic status of liver tissue in rats fed a high-lipid diet then treated with STZ.

## 4. Materials and Methods

### 4.1. Chemicals and Reagent Kits

Streptozotocin was purchased from Gentihold Biotechnology Co., Ltd. (Beijing, China). Glucose test strips were purchased from Johnson & Johnson Medical Ltd. (Shanghai, China). Curcumin was purchased from Baoji Chenguang Biotechnology Co., Ltd. (Shaanxi, Baoji, China). Sodium carboxymethylcellulose (CMC-Na) and citrate buffer were purchased from Shanghai Yuanye Biotechnology Co., Ltd. (Shanghai, China). A lipid oxidation (MDA) detection kit, total superoxide dismutase assay kit with WST-8, and BCA protein concentration assay kit (enhanced) were purchased from Biyuntian Co., Ltd. (Beijing, China). Bcl-2 antibody was purchased from Abcam plc (Cambridge, UK). Bax antibody was purchased from Cell Signaling Technology, Inc. (Danvers, MA, USA). Horseradish peroxidase (HRP) labeled goat anti-rat immunoglobulin G (IgG) was from Zhongshan Jinqiao Biotechnology Co., Ltd. (Beijing, China). TC, TG, LDL-C, HDL-C, ALT, AST, TP, and GLB detection kits were purchased from Shandong Boke Biological Industry Co., Ltd. (Shandong, Jinan, China). A 50% glucose solution was purchased from Shijiazhuang Four Drugs Co., Ltd. (Hebei, Shijiazhuang, China).

### 4.2. Experimental Animals

Sprague-Dawley rats (male) were purchased from Beijing Vital River Laboratory Animal Technology Co., Ltd., Beijing, China [SCXK (JING) 2016-0006]. The animals were housed in a temperature (22 ± 2 °C) and humidity (40–60%)-controlled area with a 12/12 h light–dark cycle. Experimental animals were acclimatized for one week before the start of the experiment. Animal care and experimentation complied with the institutional guidelines for the health and care of experimental animals. The experimental protocols were approved by the Committee on the Ethics of Animal Experiments of Nankai University (No.16/06042018).

### 4.3. Experimental Design and Procedures

In our experiment, 26 male SD rats weighing 170–190 g were randomly divided into a standard pellet diet group (Control group, six rats), a high-fat diet with streptozotocin (STZ) group (HF + STZ group, 10 rats), and a high-fat diet combined with curcumin and STZ group (HF + Cur + STZ group, 10 rats). The rats in the control group were fed a standard diet (The standard diet approximately contained 4.0% fat). The rats of the HF + STZ group were fed a high-fat diet containing 45.6% fat for 12 weeks, followed by a single tail vein injection of STZ (20 mg/kg body weight). The rats of the HF + Cur + STZ group were fed a high-fat diet and were simultaneously administered curcumin (300 mg/kg body weight) for 12 weeks, followed by a single tail vein injection of STZ.

The bodyweights of experimental rats were measured every week during the experiment. Glucose tolerance was tested before STZ injection. The fasting blood glucose (FBG) of experimental rats was measured after injection with STZ with a glucose meter from Johnson & Johnson Medical Ltd. (Shanghai, China) using a tail-vein blood sample. The rats were fasted for 12 h before the measurement of FBG. After 1 week of streptozotocin injection, blood glucose was estimated using glucose test strips. Higher than 16.7 mmol/L of blood glucose were considered to indicate type 2 diabetes mellitus (T2DM). The experimental animals were anesthetized with sodium pentobarbital (0.1 mg/g body weight), and blood samples were collected; serum was prepared by centrifugation (1200× g, 15 min) and stored at −80 °C for subsequent experiments. The liver tissues were collected simultaneously, washed with physiological saline, and weighed. Half of the liver tissue sample was used for histological examination, and the other half was frozen for biochemical examination. The ratio of tissue weight to body weight was calculated.

### 4.4. Determination of Glucose Tolerance in Experimented Rats

Each rat was fasted for 12 h before the oral glucose tolerance test. First, the FBG in each test rat was measured, then each test rat was administered glucose based on 2 g/kg body weight, and then the blood glucose was tested at 30, 60, 90, and 120 min using glucose test strips. The area under curve (AUC) was calculated according to the formula: AUC (mg/dL/h) = (BG0 + BG30) × 15/60 + (BG30 + BG60) × 15/60 + (BG60 + BG90) × 15/60 + (BG90 + BG120) × 15/60.

### 4.5. Estimation of Biochemical Parameters in the Blood×

The relative clinical biochemical parameters such as TC and TG, HDL-C, LDL-C, AST, ALT, TP, and GLB in plasma samples were measured using a biochemical analyzer (Molecular Devices MD2800, San Francisco, CA, USA) and available diagnostic kits (Shandong Boke Biological Industry Co., Ltd., Jinan, China) following the respective kit specifications.

### 4.6. Estimation of Oxidative Stress Markers and Apoptosis-Related Protein Expression in Liver Tissue

For estimation of liver oxidative stress markers, the total homogenate of the liver sample was prepared in 5% (*w*/*v*) phosphate buffer (pH 7.4) using a SCIENTZ-48 high-throughput tissue grinder (Ningbo, China). The fraction of total homogenate was centrifuged at 10,000× *g* for 20 min at 4 °C. The MDA level in tissue was measured by the lipid peroxidation MDA assay kit, following the manufacturer’s recommendations. The superoxide dismutase activity was determined by the total superoxide dismutase assay kit with WST-8 (S0131, Biyuntian Co., Ltd., Beijing, China), following the manufacturer’s recommendations. For detecting apoptosis-related protein expression in the liver, the protein concentrations in the liver sample were measured using an Enhanced BCA Protein Assay kit (P0009, Biyuntian Co., Ltd., Beijing, China). To analyze the level of the target protein, the extracted protein sample was separated by SDS polyacrylamide gel electrophoresis. The protein was transferred from electrophoresis membranes to nitrocellulose membranes. Nitrocellulose membranes were blocked with a solution of 5% nonfat milk for 2 h at 25 °C, and the membranes were incubated with primary antibodies β-actin (1:1000), Bcl-2 (1:500), and Bax (1:1000) at 4 °C overnight. The membranes were washed with Tris buffered saline Tween (TBST) three times (each 5 min), followed by incubation with the secondary antibodies at room temperature for 1 h, and three washes with TBST. Chemiluminescence was visualized using Advansta Western Bright electrochemiluminescence (Advansta, Menlo Park, CA, USA).The relative protein expression was normalized to ß-actin. The expression of Bcl-2 and Bax was observed using a Tanon 4200 fully automated chemiluminescence image analysis system (Shanghai Tian Neng Technology Co., Ltd., China), and the expression density was analyzed with Image-Pro Plus 6.0 (IPP) software.

### 4.7. Histological Change Assessment in the Liver

The liver samples from experimental rats were fixed in 4% paraformaldehyde, and then paraffin sections were made. Sections approximately 5-μm-thick were stained with hematoxylin and eosin to observe histological changes in the liver under light microscopy.

### 4.8. Statistical Analysis

SPSS 20.0 (IBM, Chicago, IL, USA), and Origin 8.5 (Origin Lab Corp., Northampton, MA, USA) were utilized for data analysis. The data were presented as the mean ± standard error (SEM). Comparisons between multiple groups were performed using one-way analysis of variance (one-way ANOVA), followed by the least significant difference (LSD) test. Results with *p* < 0.05 were considered statistically significant; results with *p* < 0.01 were considered extremely significant.

## 5. Conclusions

In summary, pre-treatment with curcumin can prevent hyperglycemia and hyperlipidemia induced by high-fat diet combined with STZ treatment in rats, as well as the liver damage induced by hyperglycemia and hyperlipidemia. The underlying mechanism might involve anti-oxidative stress and anti-apoptotic effects in the liver. These data suggest that curcumin has some pharmacological significance in the control of hyperglycemia and hyperlipidemia. However, further clinical research is required in the future.

## Figures and Tables

**Figure 1 molecules-25-00271-f001:**
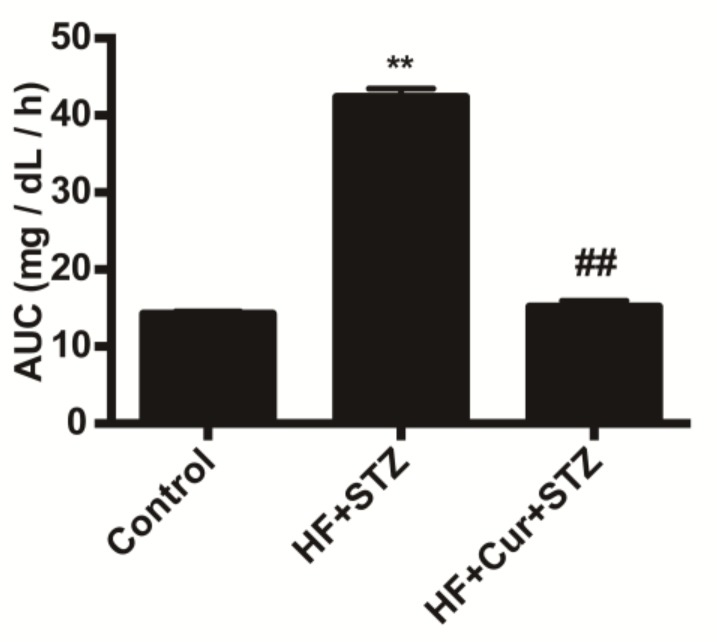
Effect of curcumin intervention on the area-under-the-curve (AUC) in rats fed a high-fat diet. HF + Cur + STZ: high-fat diet combined with curcumin and STZ group; HF + STZ: high-fat diet and streptozotocin group. The data are expressed as the mean ± SEM. ** *p* < 0.01, compared with the control group; ## *p* < 0.01, compared with the HF + STZ group; Control (*n* = 6), HF + STZ (*n* = 10), and HF + Cur + STZ (*n* = 10).

**Figure 2 molecules-25-00271-f002:**
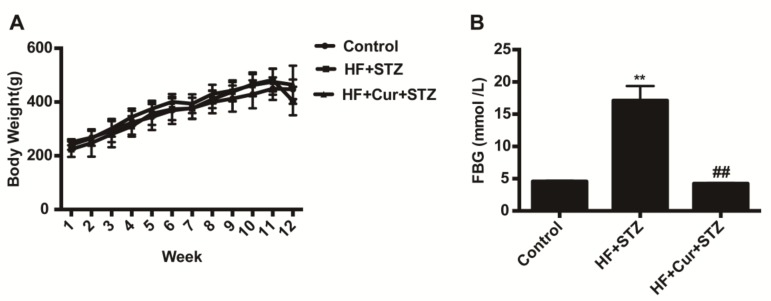
Effect of curcumin intervention on bodyweights and fasting blood glucose levels in rats fed a high-fat diet then treated with streptozotocin. (**A**) The effect of curcumin intervention on body weight. (**B**) The effect of curcumin intervention on fasting blood glucose. The data are expressed as the mean ± SEM. ** *p* < 0.01, compared with the control group; ## *p* < 0.01, compared with the HF + STZ group; Control (*n* = 6), HF + STZ (*n* = 10), and HF + Cur + STZ (*n* = 10).

**Figure 3 molecules-25-00271-f003:**
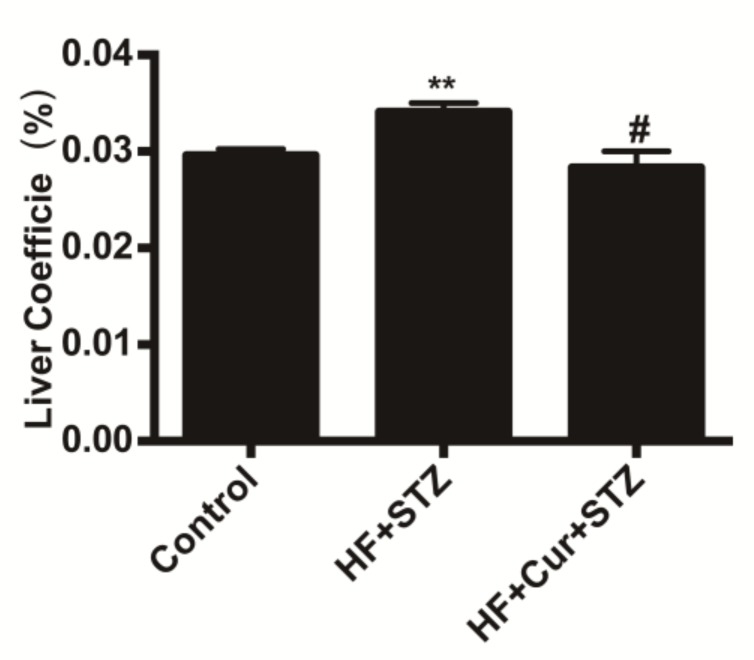
The effect of curcumin intervention on the coefficients of liver tissue to body weight in rats fed a high-fat diet then treated with streptozotocin. The data are expressed as the mean ± SEM. ** *p* < 0.01, compared with the control group; # *p* < 0.05, compared with the HF + STZ group. Control (*n* = 6), HF + STZ (*n* = 10), and HF + Cur + STZ (*n* = 10).

**Figure 4 molecules-25-00271-f004:**
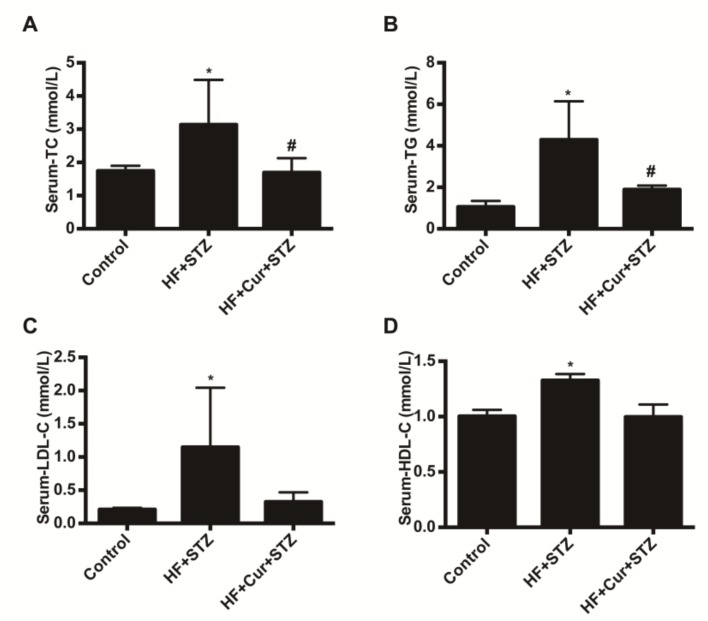
The effect of curcumin intervention on blood lipid levels in rats fed a high-fat diet then treated with streptozotocin. (**A**) The effect of curcumin intervention on total cholesterol (TC). (**B**) The effect of curcumin intervention on triglycerides (TG). (**C**) The effect of curcumin intervention on high-density lipoprotein cholesterol (HDL-C). (**D**) The effect of curcumin intervention on low-density lipoprotein cholesterol (LDL-C). The data are expressed as the mean ± SEM. * *p* < 0.05 compared with the control group; # *p* < 0.05 compared with the HF + STZ group. Control (*n* = 6), HF + STZ (*n* = 10), and HF + Cur + STZ (*n* = 10).

**Figure 5 molecules-25-00271-f005:**
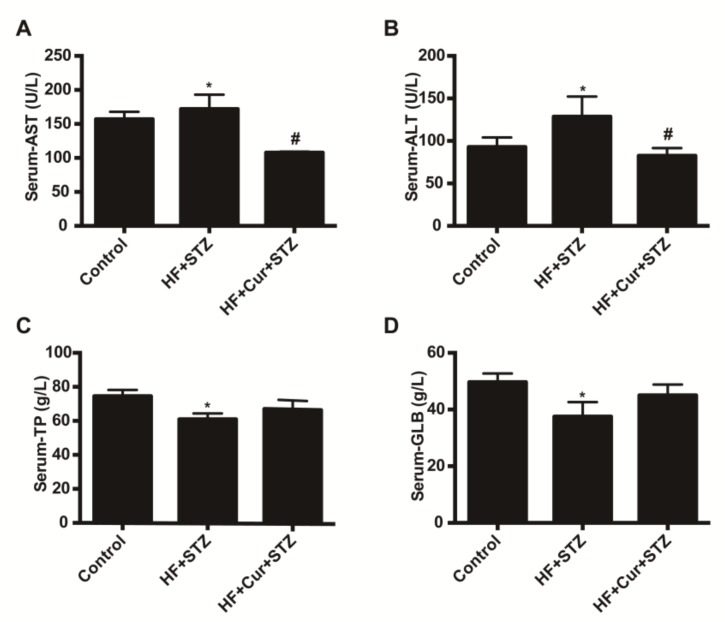
The effect of curcumin intervention on liver function in rats fed a high-fat diet then treated with streptozotocin. (**A**) The effect of curcumin intervention on aspartate aminotransferase. (**B**) The effect of curcumin intervention on alanine aminotransferase. (**C**) The effect of curcumin intervention on total protein. (**D**) The effect of curcumin intervention on globulin. The data are expressed as the mean ± SEM. * *p* < 0.05 compared with the control group; # *p* < 0.05 compared with the HF + STZ group. Control (*n* = 6), HF + STZ (*n* = 10), and HF + Cur + STZ (*n* = 10).

**Figure 6 molecules-25-00271-f006:**
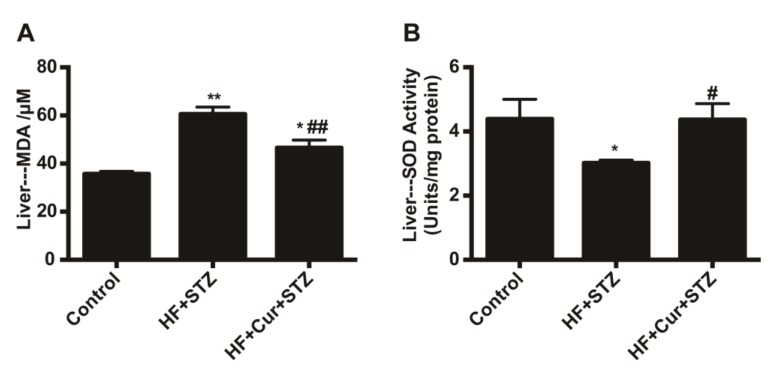
The effects of curcumin intervention on oxidative stress in rats fed a high-fat diet then treated with streptozotocin. (**A**) The effect of curcumin intervention on malondialdehyde (MDA) content in the liver. (**B**) The effect of curcumin intervention on superoxide dismutase (SOD) content in the liver. The data are expressed as the mean ± SEM. * *p* < 0.05, ** *p* < 0.01, compared with the control group; # *p* < 0.05, ## *p* < 0.01 compared with the HF + STZ group. Control (*n* = 6), HF + STZ (*n* = 10), and HF + Cur + STZ (*n* = 10).

**Figure 7 molecules-25-00271-f007:**
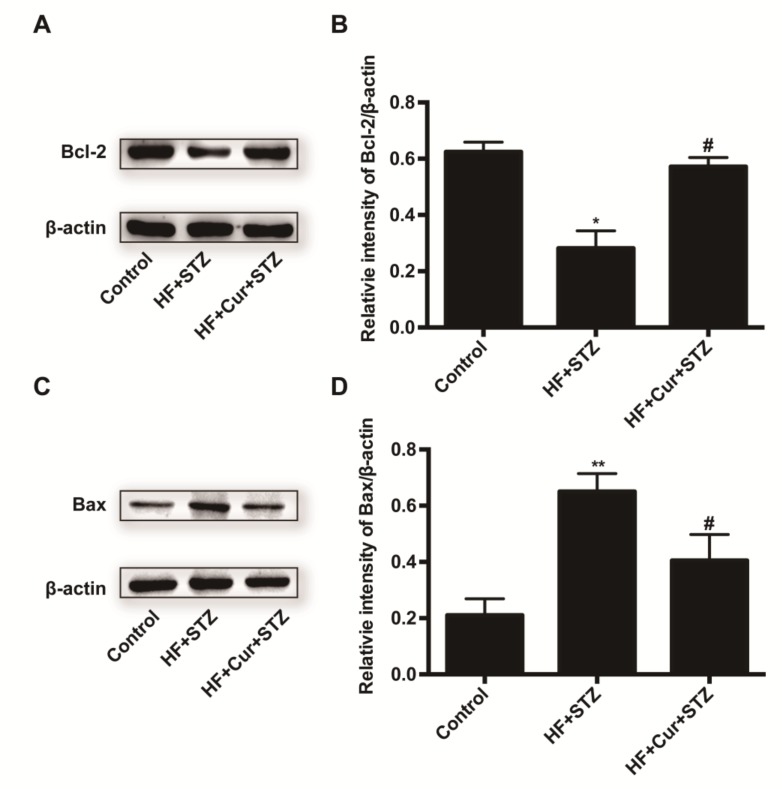
The effect of curcumin intervention, Bcl-2, and Bax expression in rats fed a high-fat diet then treated with streptozotocin. (**A**) The expression of Bcl-2 in each group after curcumin pretreatment was verified by Western blotting. (**B**) The statistical analysis was performed for the Western blotting results depicted in Figure 7A using Image Pro Plus software. (**C**) The expression of Bax in each group after curcumin pretreatment was verified by Western blotting. (**D**) Statistical analysis was performed for the Western blotting results depicted in Figure 7C using Image Pro Plus software. The data are expressed as the mean ± SEM. * *p* < 0.05, ** *p* < 0.01, compared with the control group; # *p* < 0.05, compared with the HF + STZ group. Control (*n* = 6), HF + STZ (*n* = 10), and HF +Cur +STZ (*n* = 10).

**Figure 8 molecules-25-00271-f008:**
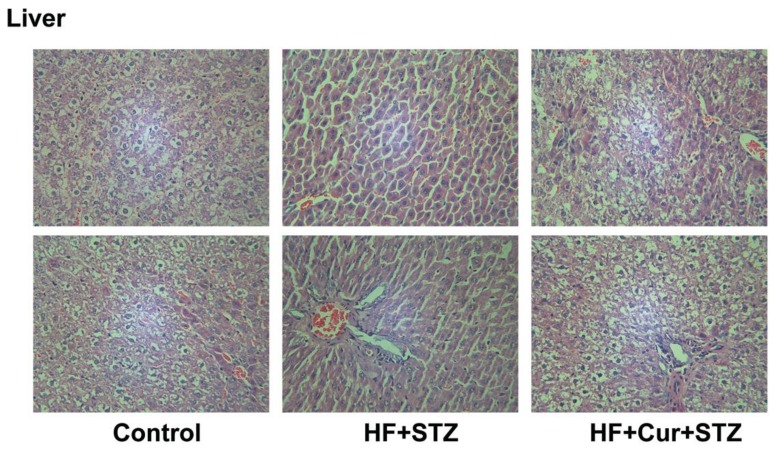
The effect of curcumin intervention on histopathological changes in the livers of rats fed a high-fat diet then treated with streptozotocin.

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
