# Peer review of "The Underlying Mechanisms of Curcumin Inhibition of Hyperglycemia and Hyperlipidemia in Rats Fed a High-Fat Diet Combined With STZ Treatment"

_molecules, 2020, doi:10.3390/molecules25020271_

Round 1

Reviewer 1 Report

The manuscript entitled “The underlying mechanisms of curcumin inhibition of hyperglycemia and hyperlipidemia in rats fed high-fat diet or a high-fat diet combined with STZ treatment” examines the effects of curcumin in anti-oxidation related protein and anti-apoptotic gene expression in the rats fed high-fat diet. Curcumin administration inhibits not only blood glucose level, but also serum triglyceride and LDL in the rats fed with high-fat diet. Moreover, SOD activity in liver of the rats fed with high-fat diet was increased by administration of curcumin. Anti-apoptotic genes such as bcl-2 and baxwere upregulated by the administration of curcumin in rats fed high-fat diet with streptozotocin treatment. The findings suggest that the curcumin is a potential compound for inhibiting onset of diabetes.

The study provides beneficial insights to use the curcumin for treatment of type II diabetes.

Comments:

1). Why did the authors test the expression of anti-apoptotic genes? What is the relationship to the inhibitory effects of curcumin for reducing blood glucose level and serum triglyceride. The rationale should be described in Introduction and/or result section. The explanation of the relationship should be discussed.

2). The authors should be checked the English in whole manuscript. For example, in abstract section, Curcuma longa (Page 1, lane 12) should be written in Italic.

3).Explanation of MDA should be described. Several readers do not know the function of MDA, therefore it is difficult to understand that content of MDA is related to antioxidation.

Author Response

Reviewer #1: The manuscript entitled “The underlying mechanisms of curcumin inhibition of hyperglycemia and hyperlipidemia in rats fed high-fat diet or a high-fat diet combined with STZ treatment” examines the effects of curcumin in anti-oxidation related protein and anti-apoptotic gene expression in the rats fed high-fat diet. Curcumin administration inhibits not only blood glucose level, but also serum triglyceride and LDL in the rats fed with high-fat diet. Moreover, SOD activity in liver of the rats fed with high-fat diet was increased by administration of curcumin. Anti-apoptotic genes such as bcl-2 and bax were upregulated by the administration of curcumin in rats fed high-fat diet with streptozotocin treatment. The findings suggest that the curcumin is a potential compound for inhibiting onset of diabetes.

Response:Thank you very much for the expression.

Comments:

1. Why did the authors test the expression of anti-apoptotic genes? What is the relationship to the inhibitory effects of curcumin for reducing blood glucose level and serum triglyceride. The rationale should be described in Introduction and/or result section. The explanation of the relationship should be discussed.

Response: We have added some content in the introduction for trying to clarify why test the expression of anti-apoptotic genes, and what is the relationship to the inhibitory effects of curcumin for reducing blood glucose level and serum triglyceride. We also have given some discussion on the question.

2. The authors should be checked the English in whole manuscript. For example, in abstract section, Curcuma longa (Page 1, lane 12) should be written in Italic.

Response: We have checked whole manuscript and change the inappropriate parts.

3. Explanation of MDA should be described. Several readers do not know the function of MDA, therefore it is difficult to understand that content of MDA is related to anti-oxidation.

Response: We have added explanation of MDA in the discussion part.

Reviewer 2 Report

The authors wrote a manuscript entitled “The underlying mechanisms of curcumin inhibition of hyperglycemia and hyperlipidemia in rats fed high-fat diet or a high-fat diet combined with STZ treatment”. The general objection refers to misleading in the title, in the aims of the study, and the conclusion.

The authors wrote:

in the title “….rats fed high-fat diet or a high-fat diet combined with STZ treatment” in the aims of the study “…Further, we also evaluated oxidative stress markers and apoptosis-related protein expression in the liver, and performed histological evaluation of the liver, in high-fat diet or a high-fat diet combined with STZ-treated rats…” in conclusion “In summary, pre-treatment with curcumin can prevent hyperglycemia and hyperlipidemia induced by a high-fat diet or a high-fat diet combined with STZ treatment in rats…”

Since there are no results of the groups which are necessary to draw such a conclusion present (e.g. rats fed with a high-fat diet and a high-fat diet + curcumin), I could not support the publication of the manuscript at the present form.

Author Response

Responses to reviewer2’ comments

Reviewer #2: The authors wrote a manuscript entitled “The underlying mechanisms of curcumin inhibition of hyperglycemia and hyperlipidemia in rats fed high-fat diet or a high-fat diet combined with STZ treatment”. The general objection refers to misleading in the title, in the aims of the study, and the conclusion.

The authors wrote:

in the title “….rats fed high-fat diet or a high-fat diet combined with STZ treatment” in the aims of the study “…Further, we also evaluated oxidative stress markers and apoptosis-related protein expression in the liver, and performed histological evaluation of the liver, in high-fat diet or a high-fat diet combined with STZ-treated rats…” in conclusion “In summary, pre-treatment with curcumin can prevent hyperglycemia and hyperlipidemia induced by a high-fat diet or a high-fat diet combined with STZ treatment in rats…”

Since there are no results of the groups which are necessary to draw such a conclusion present (e.g. rats fed with a high-fat diet and a high-fat diet + curcumin), I could not support the publication of the manuscript at the present form.

Response: Thank you very much for your advices. Based on your opinion, we have modified the description on the issue in some parts including the Title, Abstract, Results, and Materials and Methods. In fact, we hope to clarify that pre-treatment with curcumin can prevent hyperglycemia and hyperlipidemia induced by high-fat diet combined with STZ treatment, while in the rats fed with high-fat diet shown hyperglycemia and hyperlipidemia.

Reviewer 3 Report

I have reviewed the paper entitled "The underlying mechanisms of curcumin inhibition of hyperglycemia and hyperlipidemia in rats fed high-fat diet or a high-fat diet combined with STZ treatment" by Zhen-hong and colleague assessing that  curcumin prevents high-fat diet and STZ-induced hyperglycemia and hyperlipidemia, mainly via anti-oxidant and anti-apoptotic mechanisms in the liver. 

The paper could be interesting but this reviewer has some important concerns:

From the M&M section seems to understand that rats are treated with STZ every day!!!! Is this possible? The usual induction of diabetes is performing one single STZ injection in the tail vein. Authors must explain.

Moreover, the control groups HF alone, STZ alone and Curc alone have to be included in the experimental design. 

Author Response

Responses to reviewer3’ comments

Reviewer #3: I have reviewed the paper entitled "The underlying mechanisms of curcumin inhibition of hyperglycemia and hyperlipidemia in rats fed high-fat diet or a high-fat diet combined with STZ treatment" by Zhen-hong and colleague assessing that curcumin prevents high-fat diet and STZ-induced hyperglycemia and hyperlipidemia, mainly via anti-oxidant and anti-apoptotic mechanisms in the liver.

Comments:

From the M&M section seems to understand that rats are treated with STZ every day!!!! Is this possible? The usual induction of diabetes is performing one single STZ injection in the tail vein. Authors must explain.

Response: We are very sorry for not describing the problem clearly. We performed only one single STZ injection in the tail vein of rats after high-fat diet or high-fat diet combined with curcumin for 12 weeks. We have modified some description in the materials and methods section.

Moreover, the control groups HF alone, STZ alone and Curc alone have to be included in the experimental design. 

Response: Indeed, the control groups including HF alone, STZ alone and Curc alone might understand more questions. But in this experiment, we designed HF +STZ  treatment (according to many literatures) to establish DM model group, the HF +Cur +STZ group as curcumin treatment group, we hope to understand the significance of Cur preventing DM development.  

Round 2

Reviewer 2 Report

The authors made some improvements of the manuscript, but some additional clarification is needed.

1. The discussion could be more informative. Authors should try to better explain their observations.

For example:

Page 15 – Authors suggested that curcumin increases tolerance to glucose in rats fed a high-fat diet. Can they provide some possible explanation how? Page 15, lines 256-258- Authors state: „Hyperlipidemia is caused by disorders of lipid metabolism or abnormal transport in the fasting state, and shows no less than one higher-than-normal blood levels of TC, TG, and LDL-C, and is often accompanied by decreased blood HDL-C[34]“ So, how can authors explain that group with HF+STZ has the highest HDL-C? Page 18. „Generally, apoptosis is regulated by both Bax and Bcl-2 genes, and the ratio of Bax and Bcl-2 genes determines apoptosis status [51].“ It is not the ratio of genes that determines apoptosis.

2. In Results, page 13, lines 209-210 – What do the authors mean by stating „The experiment was repeated three times“?

3. In Materials and Methods authors state that 26 rats were divided into 3 groups: ctrl (n=6); HF+STZ (n=10) and HF+Curc+STZ (n=10). Yet, in Figure legends they state n=6 after the high-fat diet group? Did they exclude some of the rats from the groups? If yes, why? If not, they should clearly specify in figure legends n for each group.

4. Materials and Methods should be more informative. Some of the assays, antibodies used (e.g. MDA assay, antibodies, BCA protein assay,...) lack information about manufacturers which is important since authors mostly state that the protocols are done by manufacturer's recommendation and not specifying specifically how.

5. Page 5, lines 96-97 - abbreviations TC, TG,... need full names

6. In the majority of Figure legends fonts seem to vary in size between parts of the text.

Author Response

Responses to reviewers’ comments

Reviewer #2: The authors made some improvements of the manuscript, but some additional clarification is needed.

Comments:

The discussion could be more informative. Authors should try to better explain their observations.

For example:

Page 15 – Authors suggested that curcumin increases tolerance to glucose in rats fed a high-fat diet. Can they provide some possible explanation how? Page 15, lines 256-258- Authors state: „Hyperlipidemia is caused by disorders of lipid metabolism or abnormal transport in the fasting state, and shows no less than one higher-than-normal blood levels of TC, TG, and LDL-C, and is often accompanied by decreased blood HDL-C[34]“ So, how can authors explain that group with HF+STZ has the highest HDL-C?  Page 18. „ Generally, apoptosis is regulated by both Bax and Bcl-2 genes, and the ratio of Bax and Bcl-2 genes determines apoptosis status [51].“ It is not the ratio of genes that determines apoptosis.

Response: We have added some words in the discussion try to give rational explanation on your provided question.

In Results, page 13, lines 209-210 – What do the authors mean by stating “The experiment was repeated three times?

Response: In fact, “The experiment was repeated three times” only indicated that “the experiment of detecting the expression of Bcl-2 and Bax by western blotting repeated three times”. It might be misunderstanding expression putting it there, so we have modified the legend.

In Materials and Methods authors state that 26 rats were divided into 3 groups: ctrl (n=6); HF+STZ (n=10) and HF+Curc+STZ (n=10). Yet, in Figure legends they state n=6 after the high-fat diet group? Did they exclude some of the rats from the groups? If yes, why? If not, they should clearly specify in figure legends n for each group.

Response: We are sorry for these mistakes. Now we have given different n of each group in figure legends.

Materials and Methods should be more informative. Some of the assays, antibodies used (e.g. MDA assay, antibodies, BCA protein assay,...) lack information about manufacturers which is important since authors mostly state that the protocols are done by manufacturer's recommendation and not specifying specifically how.

Response: We have added the information on manufacturers of some reagents.

Page 5, lines 96-97 - abbreviations TC, TG,... need full names.

Response: Because we have provided full names of abbreviations TC, TG,... in the abstract, so we haven’t given full names of abbreviations TC, TG,... afterward. Now based on your advice, we have given these full names again in result section.

In the majority of Figure legends fonts seem to vary in size between parts of the text.

Response: We have adjusted the fonts of figure legends to be consistent with the text.

Reviewer 3 Report

The paper has been a bit improved but this reviewer still have a very few concerns in the description of STZ animal procedure.

1) pag 19 line 346. The sentence "followed by tail vein injection of STZ" must be changed in "followed by a single tail vein injection of STZ"

2) pag 19 line 348. the sentence  "...... followed by intravenous administration of STZ" must be changed in "... after the intravenous administration of STZ".

Author Response

Reviewer #3: The paper has been a bit improved but this reviewer still have a very few concerns in the description of STZ animal procedure.

Comments:

pag 19 line 346. The sentence "followed by tail vein injection of STZ" must be changed in "followed by a single tail vein injection of STZ"

Response: According to your advice, we have modified the sentences.

pag 19 line 348. the sentence "...... followed by intravenous administration of STZ" must be changed in "... after the intravenous administration of STZ".

Response: According to your advice, we have modified the sentence.
